

Keywords: ice nucleation, surface tension, agricultural soils, organic, macromolecules, lignin, Snomax

# The role of surface-active macromolecules in the ice nucleating ability of lignin, Snomax, and agricultural soil extracts

Kathleen A. Thompson[1,2], Paul Bieber[2], Anna J. Miller[3], Nicole Link[2], Benjamin J. Murray[1], and
Nadine Borduas-Dedekind[2]

[1]School of Earth and Environment, University of Leeds, Leeds, UK
[2]Department of Chemistry, University of British Columbia, Vancouver, Canada, V6T 1Z1
[3]Institute for Atmospheric and Climate Science, ETH Zurich, 8006 Switzerland

**Correspondence:** Kathleen Thompson (k.a.thompson@leeds.ac.uk) and Nadine Borduas-Dedekind (borduas@chem.ubc.ca)

**Abstract.**

Organic matter in agricultural soil dust can enhance dust's ice-nucleating ability relative to mineral dust, and thus impact local cloud formation. But how is this organic matter able to nucleate ice? We hypothesised that hydrophobic interfaces, such as the air-water interface, influence how organic matter nucleates ice, which can be quantified by measuring surface tension. Here, we investigated correlations between surfactant properties and ice-nucleating activities of amphiphilic macromolecules common in agricultural soils and known ice nucleators, namely lignin and macromolecules from Snomax. Lignin and Snomax solutions were analysed using our droplet freezing technique, FINC, and using an optical contact angle tensiometer. Results showed that lignin and Snomax solutions of increasing concentrations had increasing ice-nucleating activity and decreasing surface tension. In addition, high-speed cryo-microscopy of the same solutions revealed a preference for freezing at the air-water interface, consistent with these proxies being ice-active surfactants preferentially residing at the air-water interface, and thus hydrophobic surfaces. We then tested this relationship in field-collected agricultural soil extracts from the UK and Canada. Despite the trend observed for lignin and Snomax, there was no clear correlation between surface tension and freezing temperature of the soil extracts. This discrepancy may arise from the high complexity of the soil solutions, where hydrophobic interfaces in the bulk potentially compete with the air-water interface. Overall, we present further evidence of the role of hydrophobic interfaces in the heterogeneous ice nucleation of organic aerosols with implications for aerosol-cloud interactions.

## 1   Introduction

Atmospheric ice formation influences the reflectivity and longevity of clouds (Murray et al., 2012; Storelvmo et al., 2015). A small fraction of aerosols act as ice-nucleating particles (INPs), which trigger the formation of ice in clouds, creating an important mechanism for controlling cloud radiative effects and climate (Lohmann and Feichter, 2005; Ceppi et al., 2017).





However, it remains difficult to predict the physicochemical properties that influence the ability of these particles to nucleate ice.

While desert dust has been identified as a dominant source of global dust emissions (Hoose et al., 2010; Vergara-Temprado et al., 2017; Herbert et al., 2024), agricultural soil dust contributes up to 25% of the global dust emissions budget (Ginoux et al., 2012), making it a potentially important source of INPs to the atmosphere. Ice-active entities present both on leaf surfaces (Georgakopoulos and Sands, 1992; Morris et al., 2013; Hill et al., 2014) and within soils (Hill et al., 2016) in agricultural regions can be injected into the atmosphere either by passive wind erosion or by harvesting activities. Work by Lighthart (1984) found that harvesting activities increased bacteria concentrations from 643 colony-forming units m$^{-3}$ to 6,660 colony-forming units m$^{-3}$. This increase in biological aerosol particles leads to the rise in INP concentrations downwind of these harvesting activities (Suski et al., 2018). Furthermore, soil dust can be washed into nearby rivers and lakes (Knackstedt et al., 2018). These soil particles can then be aerosolised by wind and wave action at the water's surface, forming aerosols enriched in INPs (Axson et al., 2016; Cornwell et al., 2020). In rivers, high turbulence levels can also increase the release of INP-enriched aerosols into the atmosphere (Knackstedt et al., 2018).

Agricultural soils are rich in biological materials, such as bacteria, pollen, and fungi, as well as in organic matter such as macromolecules including lignin, hemicellulose, cellulose, and plant proteins, such as Rubisco (Huang et al., 2021; Alsante et al., 2023). Agricultural soil dust aerosols have an enhanced ice-nucleating ability compared to mineral dust aerosols due to the presence of biogenic material (Conen et al., 2011; Hiranuma et al., 2011; Steinke et al., 2016; O'Sullivan et al., 2016; Hiranuma et al., 2021; Pereira et al., 2022) and may contribute substantially to the global INP population (Herbert et al., 2024).

Biological material is a potential source of ice-nucleating macromolecules (INMs) in the atmosphere. Surface active compounds, also known as surfactants, are amphiphilic macromolecules, with both hydrophobic and hydrophilic moieties (Rosen and Kunjappu, 2012). Many organic macromolecules, such as fatty acids, have surfactant properties which make them important for atmospheric processes such as cloud droplet formation (Gérard et al., 2016, 2019) and these properties may also play a role in their ice-nucleating ability. Due to their amphiphilic nature, surfactant molecules partition to the surface of atmospheric aerosol droplets, reducing the surface tension of growing cloud droplets at the air-water interface Bieber and Borduas-Dedekind (2024). Therefore, surfactants lessen the barrier for further droplet growth from the condensation of water vapour onto the aerosol droplet, increasing the efficiency of cloud droplet formation (Gérard et al., 2016; Ovadnevaite et al., 2017). These hydrophobic interactions at the air-water interface of cloud droplets could also impact atmospheric ice nucleation since the location of the ice-nucleating material within the droplet has been shown to influence its ice-nucleating ability (Fornea et al., 2009).

On solid aerosol particles, surfactant coatings, such as fatty acids and alcohols, interfere with ice-active sites to either enhance ice nucleation (Hiranuma et al., 2013; Kupiszewski et al., 2016; China et al., 2017; DeMott et al., 2018) or inhibit it (Kuwabara et al., 2014; Boose et al., 2019). In aqueous droplets, surfactant macromolecules form a monolayer at the air-water interface of a droplet (Nesměrák and Němcová, 2006; Rosen and Kunjappu, 2012). Fornea et al. (2009) found that ice-nucleating substances tend to freeze at warmer temperatures when placed at the air-water interface of droplets instead of being immersed within the droplet bulk. Bieber and Borduas-Dedekind (2024) also demonstrated that freezing of ice-nucleating proteins from Snomax



occurs preferentially at the air-water interface of droplets than the bulk solution. These findings suggest that the partitioning of surfactant molecules to the droplet air-water interface could enhance the ice-nucleating ability of these substances.

Furthermore, at higher concentrations, when the surfactant macromolecules become saturated at the surface of droplets, they
begin to aggregate and partition to the hydrophobic interface and out of the aqueous solution (Rosen and Kunjappu, 2012). This behaviour results in the formation of concentration-dependent aggregates, called micelles, which could create new nucleation sites or block pre-existing sites (Gavish et al., 1990; DeMott et al., 2018). The structure of micelles formed in atmospheric aerosol droplets can vary depending on the temperature, humidity, saturation, pH, and composition of the droplet itself (Pfrang et al., 2017). These observations suggest that multiple structures could form from organic macromolecules with different ice-
nucleating ability. In addition, Bogler and Borduas-Dedekind (2020) found that the ice-nucleating ability of lignin cannot be normalised to the amount of material in the solution, suggesting the formation of concentration-dependent aggregates, or micelles, with different ice nucleating abilities.

Aggregates can form in droplet solutions in ways other than surfactant saturation and micelle formation. For many organic macromolecules, their ice-nucleating ability is closely linked to their aggregation and there is usually a critical size for aggre-
gates to form to optimise their ice-nucleating ability (Dreischmeier et al., 2017; Qiu et al., 2017; Schwidetzky et al., 2023). Proteinaceous aggregates of the bacteria *Pseudomonas syringae* nucleate ice at three different temperatures depending on the size of the aggregates (Hartmann et al., 2022; Qiu et al., 2019; Lukas et al., 2022). There are three different types of bacterial ice nucleators which are classified based on the aggregate size: Class A triggers freezing at -3°C, Class B triggers freezing at -5°C, and Class C triggers freezing at -8°C (Lukas et al., 2022). Aggregation of proteinaceous INMs is likely influenced by
electrostatic and hydrophobic interactions between proteins and components of the bacterial membrane (Lukas et al., 2020; Schwidetzky et al., 2021a). The importance of aggregation has also been observed for polysaccharides in pollen washing waters, which has been shown to exhibit ice-binding properties when aggregates are smaller than 100 kDa, and only exhibit ice-nucleating properties when aggregates are larger than 100 kDa (Dreischmeier et al., 2017; Wieland et al., 2024). In more complex samples, like fertile soils, dissolved organic material can adsorb onto larger particles or reactions can occur, leading
to coagulation and formation of complex particles and aggregates (Jackson and Burd, 1998). Although aggregation sometimes forms sites for ice nucleation, it also has the potential to hide ice-active sites. McCluskey et al. (2018) found that heating organic samples sometimes increases their ice-nucleating ability. They suggested that this increase in ice-nucleating activity was due to aggregates being dissolved and redistributed within the solution, opening up ice-active sites for nucleation (McCluskey et al., 2018). Other studies have shown that organic coatings, likely surfactants, reduce the ice-nucleating ability of mineral
dust particles (Boose et al., 2019). Therefore, the pathways and drivers of aggregation for organic INMs remain elusive, and an accurate description of the ice nucleation ability of organic matter requires further investigation.

Here, we investigated the relative contribution of surfactant macromolecules to the ice-nucleating ability of agricultural soils. We first investigated the freezing and surface activities of two proxies for soil macromolecules: lignin and Snomax. Lignin is a common biopolymer found in woody plants, which is thought to make up 30% of all organic carbon in the environment
(Boerjan et al., 2003). Snomax consists of inactive *Pseudomonas syringae* bacteria typically found on plants and in soil (Wex et al., 2015) and was used in this study as a proxy for ice-active proteins in fertile soils. The protein causing Snomax to





nucleate ice up to - 2°C is termed inaZ and is located at the membrane of the bacteria (Maki and Willoughby, 1978; Wolber et al., 1986). This protein contains an ice-nucleating template pointing towards the water, and a hydrophobic end shaping its amphiphilic properties (Hartmann et al., 2022). To nucleate ice at the highest temperatures, inaZ must aggregate and form larger structures (Qiu et al., 2019). The results of our investigations into these proxies were then compared with field samples of agricultural soils collected from the UK and Canada. The goal of this study was to gain an understanding of the contribution of surfactant macromolecules to the ice-nucleating ability of anthropogenic soils, by investigating the surface tension reduction and ice-nucleating activity of soil extracts and their components.

## 2 Methods

### 2.1 Organic matter sample collection and preparation

#### 2.1.1 Lignin

A series of aqueous suspensions of lignin (471003, Batch 1, Sigma Aldrich), ranging in concentration from $10\,\mathrm{mg\,L^{-1}}$ to $2000\,\mathrm{mg\,L^{-1}}$, were prepared in glass vials and diluted using microbiology-free reagent water (W4502, Sigma Aldrich, hereafter termed *SA water*). For consistency and comparability with previous work, we used the same batch of kraft lignin as Bogler and Borduas-Dedekind (2020) and Miller et al. (2021). The carbon concentration of the produced aqueous suspensions of lignin were determined based on a carbon content of 50%, from elemental analysis completed by the supplier (Bogler and Borduas-Dedekind, 2020). The resulting carbon concentrations ranged from $5\,\mathrm{mgC\,L^{-1}}$ to $1000\,\mathrm{mgC\,L^{-1}}$.

#### 2.1.2 Snomax

An aqueous stock solution of Snomax (Snomax® International) was prepared at a concentration of $1000\,\mathrm{mg\,L^{-1}}$ in a centrifugal tube (sterile, 50 mL, Basix, Fisher Scientific) with SA water. The Snomax solution was then filtered through a $0.22\,\mathrm{\mu m}$ syringe filter (PES membrane, sterile, Merck Millipore) to remove large aggregates and cellular fragments, and to focus the investigation on ice-nucleating macromolecules (INMs). The filtered stock solution was diluted with SA water to obtain a dilution series, ranging in concentration from $0.01\,\mathrm{mg\,L^{-1}}$ to $1000\,\mathrm{mg\,L^{-1}}$.

#### 2.1.3 Soil samples collection

Soil samples were collected from three different agricultural locations in the UK and in British Columbia, Canada (Table 1 and Fig. S1): the University of British Columbia (UBC) Farm, the University of Leeds (UoL) Farm, and Rothamsted Research. At each location, soil samples were taken from crop fields with bare soil, at least 1 m from the boundary of the field. 50 mL polypropylene centrifuge tubes (Sarstedt Inc.), which were purchased sterilised, were used to sample from the top 5 cm of soil, with at least three samples collected per location. Additionally, at each location, a clean 50 mL centrifuge tube was opened and exposed to ambient air, to examine the contamination of ice-nucleating particles from handling the samples, hereafter referred to as handling blanks. Aside from location, there were also two distinct sampling periods (Table 1). The samples taken in the





UK were obtained a few weeks after harvest season, whereas the Canadian samples were taken just a couple of hours after the soils were tilled in preparation for planting the next season's crops.

For the ice nucleation analysis, the handling blanks were treated the same as the soil samples to verify that the extracted soil
solutions were above any contamination introduced during sample manipulation. All sample tubes and handling blanks were then frozen at -20 °C until analysis to stop any biological degradation of the samples, as shown in Beall et al. (2020).

**Table 1.** Overview of the collected agricultural soil samples in Canada and in the UK.

| Sample | Location | Coordinates (°N, °E) | Crop Type | Sampling Date (D/M/Y) |
| --- | --- | --- | --- | --- |
| UBC Farm | Vancouver, Canada | 49.25, -123.24 | vegetables | 31/03/2022 |
| UoL Farm | Tadcaster, UK | 53.87, -1.32 | wheat | 12/10/2022 |
| Rothamsted Research | Harpenden, UK | 51.81, -0.36 | linseed | 29/09/2022 |

### 2.1.4 Organic matter soil extraction

A sample preparation method was developed to analyse the ice nucleation activity and surface tension of the macromolecules in the soil samples. Previous soil extraction methods first dry-sieved the soil samples to 63 μm before adding deionised water
to make up a soil suspension (Tobo et al., 2014; Suski et al., 2018). To maximise the extraction of organic macromolecules into our soil extract solutions, the soil samples were first made into a suspension with water so all organic matter could be dissolved into the solution before any further extractions were completed. First, 40 mL of MilliQ water (which was chosen to maintain consistency across the two different labs) was added to 40 g of the soil samples to create a $10^6$ mg L$^{-1}$ concentration suspension. A centrifuge (Sorvall RC5B) and a fixed-angle rotor (Thermo Scientific SS-34) separated the larger soil particles
from the prepared suspension, spinning between a relative centrifugal force of 4,300 and 12,000 g for 1 h. An average cut-off particle diameter of 0.4 μm for the extracted supernatant was calculated using Stoke's Law (Gomboš et al., 2018), assuming an average soil density of 1.3 g cm$^{-3}$ (Rai et al., 2017). Following centrifugation, the supernatant was extracted and filtered through a 0.22 μm syringe filter (PES membrane, Millipore, Sigma Aldrich) to remove the larger remaining fragments. The filtered solutions were stored in airtight containers in the fridge at 4°C until they could be analysed for their ice nucleation
activity and surface tension.

## 2.2 Analysing the Surface Activity

### 2.2.1 Surface tension measurements

The surface tension of the extracted soil sample solutions were measured using a DataPhysics optical contact angle (OCA) 15 EC Tensiometer (shown in Fig. S2). The pendant drop method was used, where an electronic dosing system dispenses
a small amount of solution to form a droplet (with a volume of 24.1±1.1 (mean ± SD) μL), suspended from the tip of a



needle. The DataPhysics SCA software uses the given value for the needle's outer diameter as a reference to determine the size of the droplet. The surface tension is then calculated automatically using the DataPhysics SCA software for OCA by fitting the Laplace-Young equation to the shape of the droplet (Berry et al., 2015). For each soil extract solution, three droplets were formed and three individual surface tension measurements were taken for each droplet. Then, the mean and the standard
deviation for the measurements were calculated.

To acquire a surface tension measurement, the droplet needs time to equilibrate with the surrounding air (Gérard et al., 2016). The required equilibrium time was determined by measuring the surface tension every 30 s for a MilliQ water droplet suspended for 5 min (Fig. S3). After about two minutes, the surface tension measurements' accuracy and reproducibility improved and remained steady for the next few minutes (Fig. S3). Therefore, in this study, the droplet was left to equilibrate for 2 min
before taking a surface tension measurement. The surface tension of pure water increases linearly with decreasing temperature (Gittens, 1969) (see Fig. S4). To address the effect of temperature on surface tension, we used a Peltier temperature control unit (TPC 160, DataPhysics) to maintain a constant temperature of approximately 22 °C with an uncertainty of ±0.3 °C during this study.

### 2.2.2    Micelle formation measurements

Two methods were used to attempt to measure the critical micelle concentration (CMC) of lignin, pyrene fluorescence and conductivity Miller (2020) (Table S3). For both, sodium dodecyl sulfate (SDS), a well-known standard surfactant, was used as a reference to ensure the methods would successfully measure the CMC if micelle formation was present.

Pyrene is a fluorescent, hydrophobic molecule which can be used to probe micellar solutions because it preferentially partitions to the interior, hydrophobic region of micelles if they are present. This partitioning changes its fluorescent intensity;
maximum changes in the fluorescent intensity occur when the surfactant concentrations are at the CMC (Kalyanasundaram and Thomas, 1977). We used the pyrene method following Aguiar et al. (2003). A dilution series of SDS and lignin were prepared, with 12 concentrations each (SDS: 0 to 18 mM; lignin: 0 to 750 mgC L$^{-1}$). A 100 µM pyrene in 90:10 acetonitrile:water solution was used to spike the SDS and lignin solutions to obtain a final pyrene concentration of 2 µM. 200 µL of each solution were then pipetted in triplicates into black polystyrene 96-well plates for analysis in the Tecan Infinite® 200 PRO Microplate
Reader (Tecan, Switzerland) using an excitation wavelength of 335 nm and measured fluorescent intensity at 373 and 384 nm. Each well was measured in triplicate, and all measurements were conducted at a set temperature of 29 °C. For calculating the CMC using the pyrene 1:3 method, the ratio of the intensity at 373 nm to the intensity at 384 nm was plotted against the surfactant concentration, which should result in a sigmoidal curve, for which the inflection point is the CMC (Figure S6).

Conductivity is also a common technique for determining the CMC for ionic surfactants because as micelles form, the
conductivity of the solution increases (Nesměrák and Němcová, 2006). The advantage of conductivity measurements is that they are relatively simple, low-cost, and can be done at varying temperatures. Here, the conductivity of SDS solutions were measured for 30 concentrations ranging from 0 to 15.2 mM at 0, 5, 10, 15, and 25 °C. For lignin, conductivity measurements were taken for solutions of up to 415 mgC L$^{-1}$ at 0 and 25 °C. Measurements were made using an Oakton Waterproof CON 150 portable conductivity meter (Fisher Scientific). Samples were kept at the desired temperature by submerging them inside





an ethanol cooling bath while simultaneously stirring by hand with the conductivity probe. The conductivity measurement was recorded only when both the temperature and the conductivity reading stabilized, typically after 2-5 min. To determine the CMC, we used the method by Carpena et al. (2002). Specifically, a Boltzmann-type sigmoid curve was fit to the first derivative of the conductivity-concentration data, where the inflection point equals the CMC (Figure S7).

## 2.3 Instrumentation and Sample Analysis

### 2.3.1 Total organic carbon (TOC) analysis

The total organic carbon (TOC) analysis of the agricultural soil extracts was quantified using a high-temperature TOC analyser in the differential mode (Multi NC2100, Analytik Jena). The analyser determines the amount of total carbon (TC) by injecting $150\,\mu L$ of the sample directly into a combustion tube set at a temperature of $800°C$. The carbon is digested and TC is measured from the detection of the generated carbon dioxide by non-dispersive infrared spectrometry (NDIR). Next, the analyser

determines the total inorganic carbon (TIC) by injecting another $150\,\mu L$ of the sample into the TIC condensation vessel, where phosphoric acid is added and then the carbon dioxide is purged and detected. The software then calculates the TOC as the difference between TC and TIC.

The TOC analyser was calibrated using standards diluted from commercially prepared stocks of 1000 ppm TOC (76067-250ML-F, Merck Life Science UK Ltd.) and 1000 ppm TIC (12003-250ML-F, Merck Life Science UK Ltd) (see Table S1).

After filtration to $0.22\,\mu m$, $2\,mL$ of each soil extract solution was pipetted into glass auto-sampler vials and secured with a snap cap. A handling blank was analysed alongside the sample solutions. The difference in measured TOC between the sample and the blank was calculated to determine the organic carbon from the soil (see Table S2). TIC concentrations were up to $26.3\,mg\,C\,L^{-1}$; this small concentration will not affect the freezing point depression of the aqueous droplets.

The organic carbon content of the Snomax sample was measured using a Shimadzu TOC-LCPH instrument in the non-

purgeable organic carbon (NPOC) mode. The instrument was set to a combustion temperature of $680°C$, a sparge time of 1:50 minutes, and a gas flow of $80\,mL\,min^{-1}$. Sample acidification was 1%. Standards were prepared from a potassium phthalate TOC standard solution ($1000\,mg\,L^{-1}$, Sigma Aldrich) in concentrations of 2.0, 5.0, 10.0, and $20.0\,mg\,L^{-1}$. The calibration curve (Fig. S5) resulted in ($y = 4.999x + 1.120$, $R^2 = 0.999$). The Snomax sample was diluted by a factor of 10 prior measurements to fall within the range of the calibration curve.

### 2.3.2 High-speed cryo-microscopy measurement

To analyse the onset freezing location of the droplets, a high-speed cryo-microscopic technique was used as described in detail in Bieber and Borduas-Dedekind (2024). This technique can distinguish between freezing onsets at the air-water interface (AWI) and the bulk volume of the droplets. Briefly, a sample droplet ($0.5\,\mu L$) was pipetted onto a glass slide which was previously coated with a repellent polymer (Fluoropel 800, 0.2%, Cytonix, USA) to avoid interactions of the ice-nucleating

substances with the glass slide. Afterwards, another glass slide was placed onto the droplet, separated by $220\,\mu m$ thick spacers to create a cylindrical droplet. This assembly was cooled by -3°C/min until ice formation was observed optically. An ice





nucleation event at the AWI can be distinguished from ice nucleation in the bulk of the droplet based on statistics and the choice of optimal iterations is described in (Bieber and Borduas-Dedekind, 2024). To quantify the difference between freezing at the AWI versus in the bulk, the normalized polar coordinates for 32 freezing events were calculated and plotted as target plots. For the data analysis, the volume was divided into 5 equivalent concentric segments and the frequency of ice nucleation in each segment was calculated. Bulk nucleation would result in a frequency of 20% for each segment as shown theoretically with Monte Carlo simulations and experimentally with birch pollen samples (Bieber and Borduas-Dedekind, 2024). If a sample is ice-active at the AWI, the resulting frequency in the outermost segment will be enhanced (see Figure 2).

### 2.3.3 Ice nucleation analysis

The drop Freezing Ice Nuclei Counter (FINC) is a custom-built drop freezing technique used to investigate heterogeneous ice nucleation by immersion freezing, as described by Miller et al. (2021). Briefly, FINC consists of an ethanol bath (LAUDA Proline RP 845, Lauda-Königshofen) used to cool the sample, with a mounted camera and LED lights for detecting freezing based on changes in light intensity. Each cooling experiment consists of three 96-well Piko PCR trays, with each well containing $10\,\mu$L aliquots of sample solutions. Solutions were pipetted into the PCR trays inside a laminar flow hood to reduce contamination during sample preparation. FINC cools at a rate of $1\,^{\circ}$C min$^{-1}$ until an endpoint of -32$^{\circ}$C, and the camera takes an image every $0.2^{\circ}$C. The recorded freezing temperatures were then calibrated to account for the temperature difference within each well (see Sect. S5 in SI).

The fraction of droplets frozen as a function of temperature, $f_{\text{ice}}(T)$, was determined from the corrected freezing temperatures, as shown in Eq. (1), where $N(T)$ is the number of droplets frozen at temperature, $T$, and $N_{\text{total}}$ is the total number of droplets in the PCR tray.

$$f_{\text{ice}}(T) = \frac{N(T)}{N_{\text{total}}} \tag{1}$$

To further quantify the ice-nucleating activity, the sample solutions were normalised by their organic contents. The active site density per mass of carbon, $n_{\text{m}}$, was calculated as a function of $f_{\text{ice}}(T)$, as shown in Eq. (2); where $V_{\text{d}}$ is the volume of sample in each well ($10\,\mu$L for each experiment in this case) and $C_{\text{mC}}$ is the mass concentration of dissolved organic carbon within the sample solution. The TOC mass concentration of the soil extract solutions, $C_{\text{mC}}$ was quantified using a TOC analyser.

$$n_{\text{m}}(T) = -\frac{ln(1 - f_{\text{ice}}(T))}{V_{\text{d}}C_{\text{mC}}} \tag{2}$$

### 2.3.4 Heat treatment experiments

Heat treatment experiments were carried out using the protocol described in Daily et al. (2022). For the UBC Farm and UoL Farm soil sample extracts, $5\,$mL aliquots were transferred into $50\,$mL polypropylene centrifuge tubes (Basix, Fisher Scientific). A heating bath from a rotary evaporator (B-490, Buchi) was filled with deionized water, and a clamp stand was used to secure the centrifuge tubes in position to fully immerse the sample within the water bath at 98$^{\circ}$C for 30 mins. The tubes were tightly closed to prevent evaporation, which would lead to an increase in the solution concentration. After heating, the aliquots were left to cool before further surface tension and ice nucleation analysis.





## 3   Results and Discussion

### 3.1   Soil Proxies

#### 3.1.1   Surface tension of lignin and Snomax

First, we measured the surface tension of lignin and Snomax solutions using an OCA tensiometer as a function of concentration (Fig. 1). As expected, the greatest surface tension reductions were observed for the most concentrated solutions of both lignin and Snomax (Fig. 4a). Specifically, a $5.8\,\mathrm{mN\,m^{-1}}$ reduction was observed for the $1000\,\mathrm{mgC\,L^{-1}}$ lignin solution. Comparatively, an even larger reduction of $8.5\,\mathrm{mN\,m^{-1}}$ was observed for the Snomax mass concentration of $125.2\,\mathrm{mgC\,L^{-1}}$. The concentration of lignin and Snomax needed to exceed $200\,\mathrm{mgC\,L^{-1}}$ and $62.5\,\mathrm{mgC\,L^{-1}}$, respectively before a measurable surface tension reduction of at least $0.5\,\mathrm{mN\,m^{-1}}$ was observed (Fig. 1). We measured an average surface tension for pure water at $22°\mathrm{C}$ of $72.2\,\mathrm{mN\,m^{-1}}$ (see Fig. S3). We hypothesised that the surface-active soil proxies were accumulating at the air-water interface of the suspended droplet on the tensiometer and nucleating ice at this hydrophobic interface as demonstrated by Bieber and Borduas-Dedekind (2024). Indeed, Bieber and Borduas-Dedekind (2024) showed that the hydrophobic air-water interface was the location of freezing for ice-nucleating proteins in Snomax.

The threshold concentration of $62.5\,\mathrm{mgC\,L^{-1}}$ of Snomax was comparable to TOC concentrations found in the atmosphere. For instance, TOC concentrations of cloud water droplets measured by Cook et al. (2017) and Pratt et al. (2013) were up to $16.6\,\mathrm{mgC\,L^{-1}}$ and up to $9.9\,\mathrm{mgC\,L^{-1}}$, respectively. Furthermore, Gérard et al. (2016) collected PM$_{2.5}$ in Sweden and measured a concentration of anionic, cationic, and nonionic surfactants up to $27 \pm 6$ mM, which we estimate to be approximately equivalent to $3000\,\mathrm{mgC\,L^{-1}}$ based on an average molecular weight of 225 g mol$^{-1}$ from the range of 100-350 g mol$^{-1}$ predicted by Yazdani et al. (2021), and assuming a 50% carbon content. These observations suggest that atmospheric concentrations of environmental surfactants may reduce the surface tension of cloud droplets (Gérard et al., 2016, 2019).

#### 3.1.2   IN ability increased with a decrease in surface tension

Next, we examined the ice-nucleating ability of two common proxies of atmospherically-relevant organic macromolecules, lignin and Snomax, using the droplet freezing assay FINC (see Fig. S10 for frozen fractions). The $200\,\mathrm{mgC\,L^{-1}}$ solution of lignin had a median freezing temperature $T_{50}$ of -19.0°C, corroborating the -18.8°C reported by Bogler and Borduas-Dedekind (2020) and the -22.0°C reported by Bieber et al. (2024). Next, the lignin solution was diluted by three orders of magnitude to obtain values approaching the handling blank (Fig. S10a). The $125.2\,\mathrm{mgC\,L^{-1}}$ solution of Snomax nucleated ice at a $T_{50}$ value of -5.8°C (Fig. S10b). A dilution series along five orders of magnitude led to the characteristic step-wide frozen fraction curve of Snomax, corroborating the freezing temperatures reported by (Wex et al., 2015), which observed a range in freezing temperatures of Snomax solutions between -2°C and -9°C. Filtering the Snomax solution to $0.22\,\mathrm{\mu m}$ resulted in a small decrease in its freezing activity, particularly at -2°C (Fig. S11), indicating the loss of Class A aggregates due to filtration (Lukas et al., 2022).



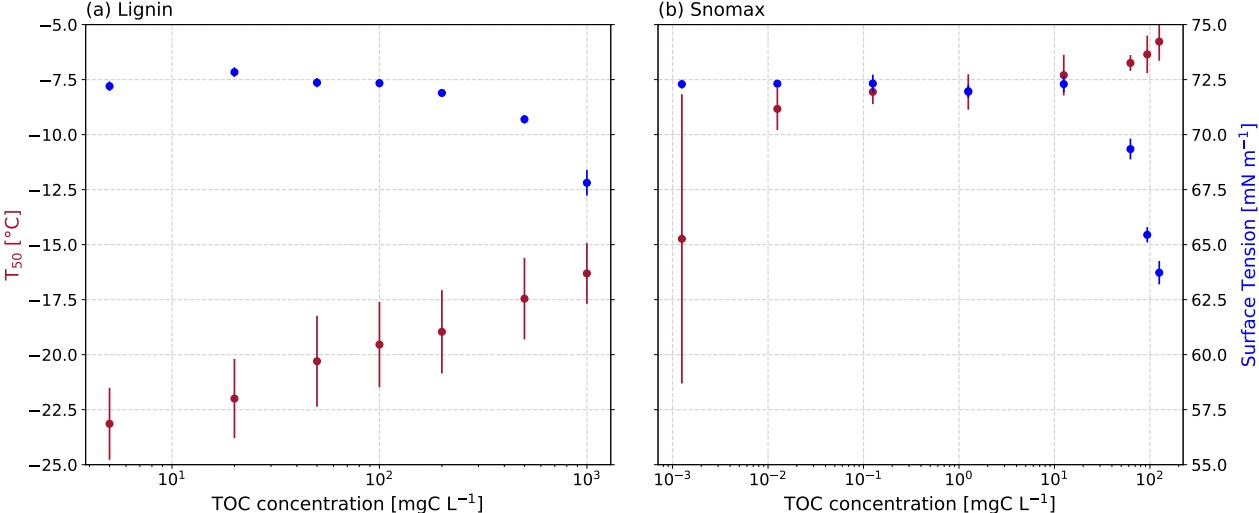

**Figure 1. Surface tension (blue) and average freezing temperature ($T_{50}$, red) plotted against the mass concentration of the analysed soil proxies.** (a) Lignin and (b) Snomax.

Increases in the concentrations of lignin and Snomax led to an observed increase in $T_{50}$ and correlated to decreases in surface tension (Fig. 4a). This observed correlation was statistically significant for both lignin and Snomax (lignin: $r^2 = 0.82$, p = 0.0136; Snomax: $r^2 = 0.79$, p = 0.0072). Surfactants have previously been shown to enhance ice-nucleating activity (Hiranuma et al., 2013; China et al., 2017; Qiu et al., 2017; DeMott et al., 2018; Perkins et al., 2020; Schwidetzky et al., 2021b). For example, some surfactant macromolecular layers are considered a good lattice match for ice formation, such as those formed from fatty acids (Qiu et al., 2017; DeMott et al., 2018). Our analysis showed that heterogeneous ice nucleation at moderate supercooling temperatures (> -9 °C) was observable with concentrations as low as $0.01\,\mathrm{mgC\,L^{-1}}$ of Snomax. Overall, the observed correlation between measured surface tension reduction and IN activity of the soil proxies indicates that the formation of a surfactant monolayer may play a role in heterogeneous ice nucleation.

Next, we can estimate the concentration of lignin required for a monolayer to form in our tensiometer droplets (see SI section 6 for equations). The lignin used in this study has a molecular weight of approximately $10^4\,\mathrm{g\,mol^{-1}}$(Bogler and Borduas-Dedekind, 2020). Assuming a density of $1\,\mathrm{g\,cm^{-1}}$ and assuming the polymer is spherical, the diameter of a single lignin polymer would be 3.2 nm. To calculate the surface area taken up by one lignin molecule, we used this average radius to calculate the average cross-sectional area of one lignin molecule to be $8.04 \times 10^{-14}\,\mathrm{cm^{-2}}$. The average volume of the droplets used in our tensiometer was 24.1 μL, making the average droplet surface area equal to $0.403\ \mathrm{cm^{-2}}$. Using these, we were able to show that the average lignin concentration required to form a monolayer in our droplets is $3.46\,\mathrm{mg\,L^{-1}}$. This concentration is much less than the maximum lignin concentration examined in this study, indicating that there was a potential that our lignin solutions were saturated at the air-water interface in our measurements.



### 3.1.3 High-speed cryo-microscopy to identify freezing at the air-water interface

We used a high-speed camera on top of a cryo-microscope to locate the onset freezing in single droplet experiments Bieber and
Borduas-Dedekind (2024). This technique identified that the onset freezing location of Snomax samples statistically favoured
the air-water interface (AWI) (Fig. 2b). In fact, Bieber and Borduas-Dedekind (2024) found the onset freezing of a filtered
Snomax sample to be at the outermost fifth of the droplet in 59% of the 32 experiments. To clarify, if freezing occurred in the
bulk, we would have expected only 20% of freezing onsets to occur at the AWI according to Monte Carlo simulations (Bieber
and Borduas-Dedekind, 2024). Here, we further tested lignin solutions in our high speed cryo-microscope, and found that the
nucleation frequency was at the AWI in 34 % of the 32 measurements (Fig. 2a). This result indicates that there is enhanced
ice-nucleating activity of lignin at the AWI (Fig. 2a), however, it is less pronounced compared to the Snomax samples (Fig.
2b).

Considering the smaller droplet volumes used for the high-speed freezing assay, which were only $0.5\,\mu L$ compared to the
$10\,\mu L$ used in FINC, we would expect the freezing activity to vary between the two techniques. In addition, the solution
concentrations for lignin and Snomax used in the high-speed freezing assay were higher ($5000\,mgC\,L^{-1}$ and $125\,mgC\,L^{-1}$,
respectively) compared to Fig. 1 which may lead to more pronounced AWI activity.

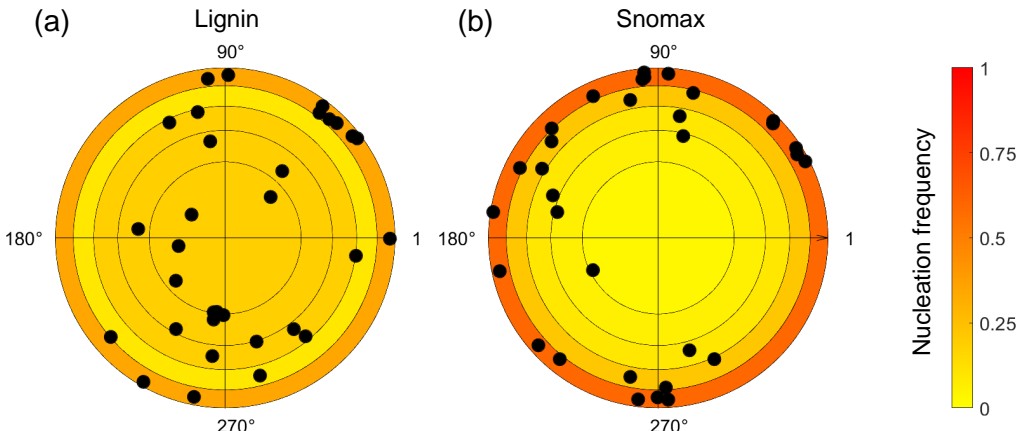

**Figure 2. Onset freezing locations for droplets squeezed between two protein-repellent glass slides** (a) Lignin ($5000\,mgC\,L^{-1}$) and
(b) literature data for filtered Snomax ($125\,mgC\,L^{-1}$) from Bieber and Borduas-Dedekind (2024). The mean freezing temperatures were
-20.6°C for lignin and -6.3°C for the filtered Snomax sample, which is in the range of the investigated concentrations. Note that the droplet
volume was $0.5\,\mu L$ for these measurements.



### 3.1.4 Formation of micelles

The observed surface tension reductions may also indicate the potential for micelle formation within the solutions (Mabrouk et al., 2022). As surfactants become saturated at the surface of a droplet, as is likely the case in this study, the macromolecules

cluster together into micelles to minimise the exposure of the hydrophobic moieties to the water (Rosen and Kunjappu, 2012). The concentration of surfactants needed to initiate micelle formation is called the critical micelle concentration (CMC) (Rosen and Kunjappu, 2012). Once micelle formation is initiated, further increases in concentration do not lead to further decreases in surface tension since the surface is saturated (Nesměrák and Němcová, 2006). We hypothesised that the formation of micelles may create or cover ice-active sites for nucleation.

To investigate the potential impact of micelle formation on the ice-nucleating activity of lignin, we used pyrene fluorescence (Aguiar et al., 2003) and conductivity (Nesměrák and Němcová, 2006) to try to measure the CMC of our lignin suspensions (Fig. S6 and S7 and Table S3). Using these techniques, we were unable to measure a CMC for lignin. Considering that our complementary measurements using sodium dodecyl sulphate (SDS) yielded a CMC for both techniques, we were able to confirm that the measurement techniques worked. Instead, the results from our lignin suspensions indicate that lignin is not

supersaturated at the AWI and that micelle formation is not occurring within these suspensions. Therefore, we can conclude that the formation of micelles was not necessary for lignin to nucleate ice and it is likely that interactions at the AWI are more important for the ice-nucleating ability of lignin, as suggested by Bieber et al. (2024).

### 3.2 Soil extracts

### 3.2.1 Surface tension of soil extracts

Next, we investigated whether the observed relationship between ice-nucleating activity and surface tension reduction also held true for field-collected agricultural soil extract solutions from the University of British Columbia (UBC) Farm, the University of Leeds (UoL) Farm and Rothamsted Research Farm (Fig. 3). Unlike the soil proxies presented above, we did not observe a relationship between the ice-nucleating ability of the soil extract solutions and their surface tension reduction: UBC: $r^2 = 0.25$, $p = 0.5032$; UoL: $r^2 = 0.22$, $p = 0.689$; Rothamsted: $r^2 = 0.01$, $p = 0.9868$ (Fig 4b).

The largest surface tension reduction observed from the soil extract solutions in this study was 2.2 mN m$^{-1}$, but this change did not correlate with TOC concentrations or freezing activity (Fig. 3). We hypothesised that the complexities of the soil solutions could be enhancing the aggregation of the organic matter thereby creating ice-active sites. Although the soil extract solutions were filtered to 0.22 μm, the solutions still consisted of a complex mixture of biological and mineral components. Most likely, the solutions were made up of fragments of organic material from plant debris, bacteria, fungi, pollen, etc., as

well as nanoscale mineral particles, such as clays (Conen et al., 2011; Steinke et al., 2016, 2020). The presence of these other molecular components within the soil solutions can act as hydrophobic surfaces for the accumulation and aggregation of ice-active entities. So, instead of forming at the AWI, aggregates can form within the bulk of the water droplets (Bieber and Borduas-Dedekind, 2024; Bieber et al., 2024). The lack of relationship between surface tension and freezing activity in the soil



solutions indicates that surfactant-like macromolecules are not required to accumulate at the AWI for the heterogeneous ice
nucleation to be observed in supercooled droplets of agricultural soil extracts.

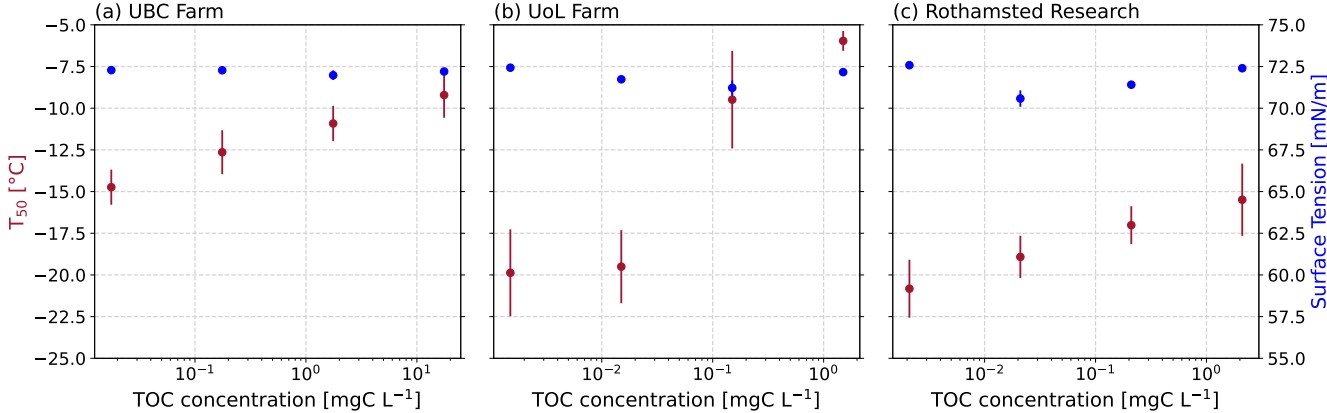

**Figure 3. Surface tension and average freezing temperature ($T_{50}$) plotted against the mass concentration of the soil extracts** from (a)
UBC Farm, (b) UoL Farm and (c) Rothamsted and each of their corresponding dilution series.

### 3.2.2 IN ability of soil extracts

The soil extract solutions displayed a range of atmospherically relevant freezing activity (Fig. 5). Specifically, the $T_{50}$ of the
undiluted soil extracts ranged from -15.5°C to -6.3°C, a difference of 11.2°C. All the freezing activities for the soil extracts
were above the handling blanks (Fig. 5), indicating that the observed freezing could be attributed to the soil samples. Our
results demonstrate the breadth of freezing ability of a range of soil extracts across two continents.

### 3.2.3 Sensitivity of soil extracts to heat treatment

To identify heat-label biological material in the soil extracts (Hill et al., 2016; O'Sullivan et al., 2018; Daily et al., 2022), heat
tests were performed on the two extracted soil dilution series from the samples with the highest ice-nucleating activities from
UoL Farm and UBC Farm. We observed a loss of the ice-nucleating activity (Fig. 6), particularly at around -5°C in the $n_{m}$
spectra, suggesting that ice-nucleating proteins are being denatured, as a result of the heat treatment (Steinke et al., 2016; Suski
et al., 2018; Daily et al., 2022; Wieland et al., 2024). In the undiluted extracted soil solution for the UoL sample 1, we observed
a shift in the $T_{50}$ from -7.2°C to -11°C and for the undiluted UBC sample 1, there was a decrease in the $T_{50}$ from -9.2 to
-10.6°C. In general, the ice-nucleating ability of the soil extracts remains high, despite exposure to heat treatment which would
reduce the freezing temperatures (Garcia et al., 2012; Suski et al., 2018). Heat-stable INMs have been previously observed
in soil samples, which break down after hydrogen peroxide treatment (O'Sullivan et al., 2014; Suski et al., 2018). Hill et al.
(2016) found a large fraction of the ice-nucleating ability of agricultural soils to be resistant to all tests except oxidation by
hydrogen peroxide. They concluded that this ice-nucleating ability was likely attributed to plant material (Hill et al., 2016).





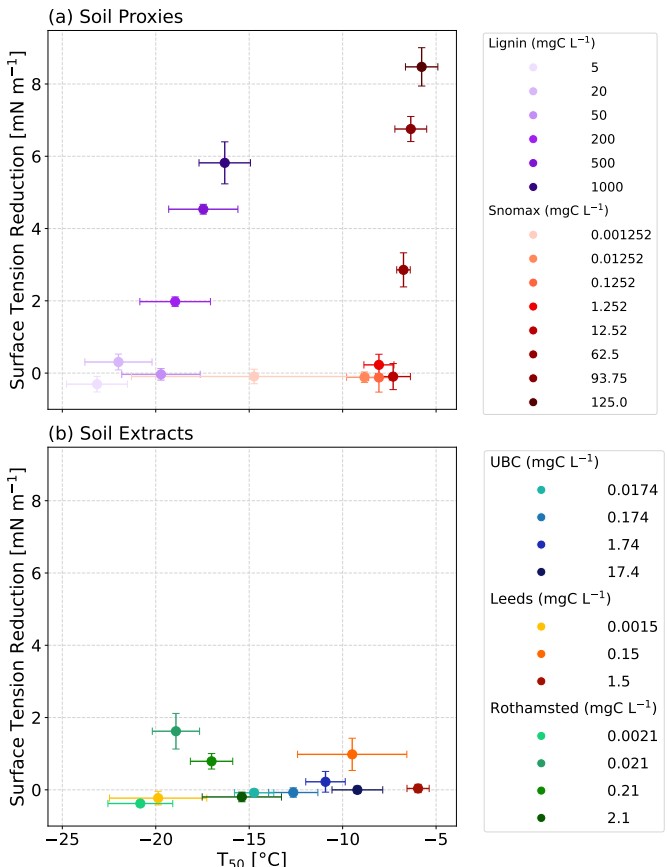

**Figure 4. The average freezing temperature ($T_{50}$) plotted as a function of surface tension for the dilution series of (a) proxy solutions and (b) our soil extracts.** For the proxies, a significant correlation was observed (lignin: $r^2$ = 0.82, p = 0.0136; Snomax: $r^2$ = 0.79, p = 0.0072). For the soil extracts, no significant correlation was observed (UBC: $r^2$ = 0.25, p = 0.5032; UoL: $r^2$ = 0.22, p = 0.689; Rothamsted: $r^2$ = 0.01, p = 0.9868).

These findings are similar to those in our study, where the freezing activity remained unaffected by the heating, suggesting that the majority of the ice-nucleating activity was attributable to heat-stable INMs from the breakdown of plant material, such as
lignin, cellulose, starch, or pectin (Borduas-Dedekind et al., 2019; Steinke et al., 2020; Chen et al., 2021).

In addition, a small positive shift in freezing temperature of $2\,^{\circ}\text{C}$ was observed in the $0.0015\,\text{mgC}\,\text{L}^{-1}$ concentration UoL extract (Fig. 6). Combined with the overall lack of a noticeable loss in freezing activity as a result of freezing, this observation suggests that ice-active entities were released during the heating of the soil samples. Similar observations have been made previously, for example by McCluskey et al. (2018), who showed that ice nucleation was enhanced after heating marine
bioaerosols. They hypothesised that this increased ice-nucleating activity was due to the breakdown of cell walls and the release of INMs into suspension. Since we filtered our soil samples to $0.22\,\mu\text{m}$, whole cells were removed from the suspension, and



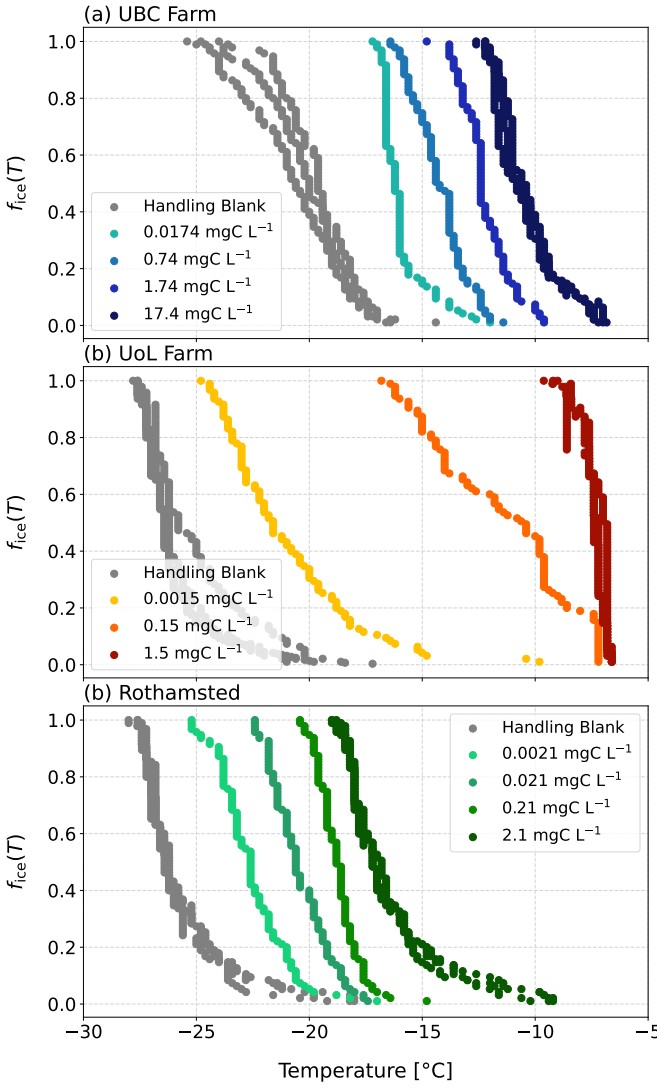

**Figure 5. Fraction frozen ($f_{ice}(T)$) curves as a function of temperature for the three soil extract solutions and their dilutions** from (a) UBC Farm, (b) UoL Farm and (c) Rothamsted. The indicated solution concentrations refer to the carbon content of each solution.

thus the breakdown of cell walls was unlikely to be a source of INMs in our study. The authors also suggested that the observed increase in ice-nucleating activity could be due to the heat causing the dissolution of aggregates in suspension, redistributing macromolecules and exposing new ice-active sites (McCluskey et al., 2018). Therefore, the observed increase in ice-nucleating

activity after heating our UoL soil extracts and the maintenance of ice-nucleating activity despite heating suggests that (a) there are aggregates present in the soil suspension which are not micelles and (b) these aggregates are inhibiting ice nucleation at temperatures below about -12°C.




Surface tension measurements were also taken before and after heating the soil extract solutions (see Table S2). We anticipated that removing heat-sensitive INMs, such as proteins, may have allowed us to see the relationship between surface

tension and ice-nucleating activity, as observed in our lignin solutions (Fig. 4a). However, for both the UoL Farm and UBC Farm samples, we observed little change in the surface tension reduction after heating (Table S2). Therefore, no relationship between the ice-nucleating ability of the soil extract solutions and their surface tension reduction was observed after heating the soil extracts.

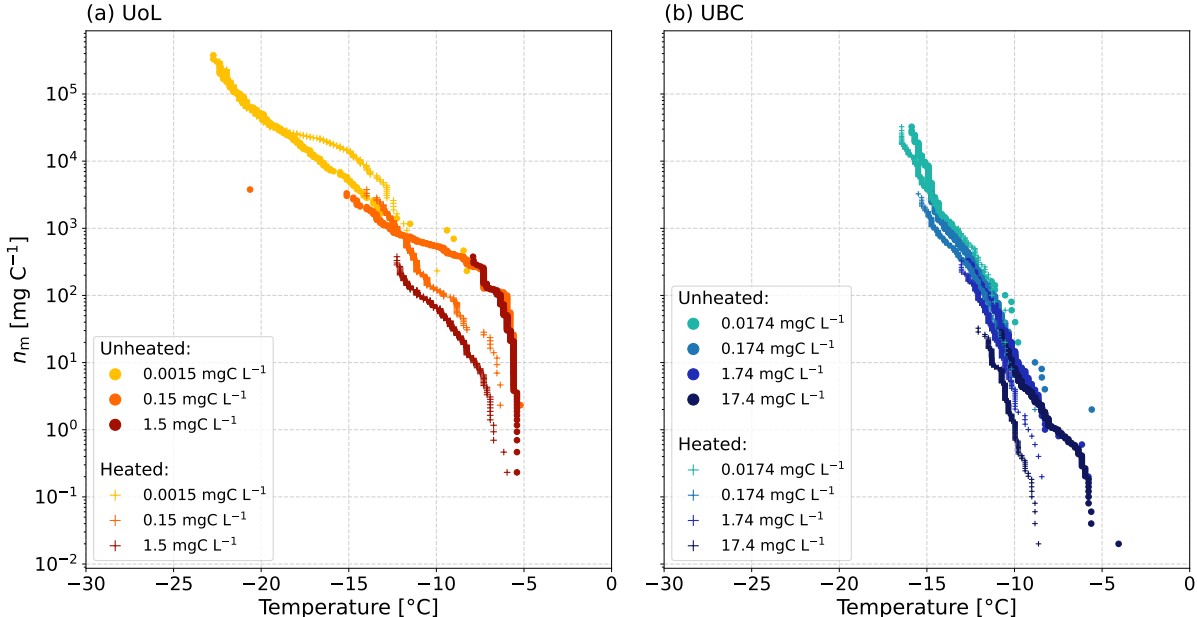

**Figure 6. Ice active mass site density ($n_{\mathrm{m}}$) as a function of the temperature of the dilution series for soil extracts before and after heat treatments.** (a) The UoL Farm and (b) the UBC Farm.

## 3.3 Normalised Freezing Spectra

Furthermore, we examined the normalised ice-nucleating activity of the lignin and Snomax dilutions series. The observed freezing activity for the lignin normalised to the carbon content ($n_{\mathrm{m}}$) aligns with previous work done by Bogler and Borduas-Dedekind (2020) and Miller et al. (2021) (Fig. 7a). The $n_{\mathrm{m}}$ tended to increase with decreasing concentration of lignin. This trend was also observed by Bogler and Borduas-Dedekind (2020) and is further demonstrated by the parameterisation by Miller et al. (2021), which indicates higher $n_{\mathrm{m}}$ for the lower concentrations of lignin ($20\,\mathrm{mgC\,L^{-1}}$). The aggregation of lignin molecules

at higher concentrations may obscure ice-active sites. Indeed, once the aggregates are dissolved at lower concentrations, the ice-nucleating activity increases as the ice-active sites are released into solution.





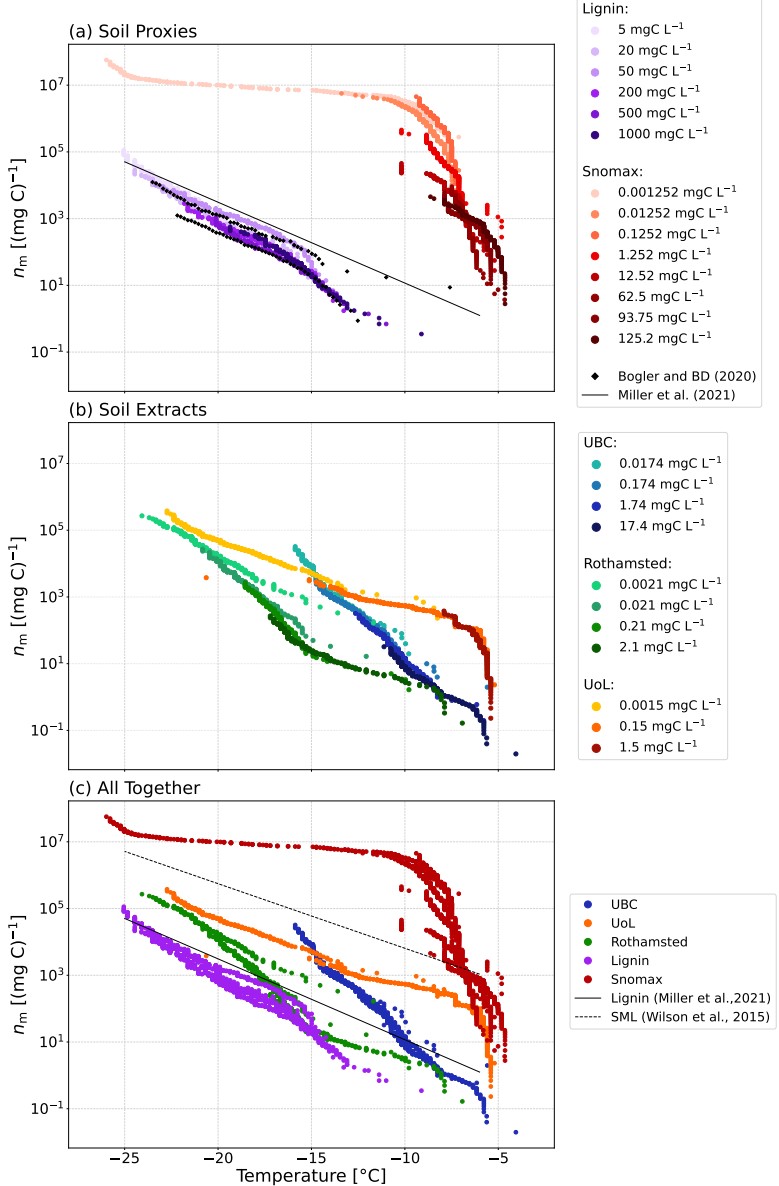

**Figure 7. Ice-active mass site density ($n_{\mathrm{m}}$) as a function of temperature for** (a) the soil proxies, lignin and Snomax, and their dilutions, (b) the three soil extracts and their dilutions from the UBC Farm (Vancouver, Canada), the UoL Farm (Tadcaster, UK) and Rothamsted (Harpenden, UK), and (c) the soil proxies, soil extracts compared together. For comparison, data from Bogler and Borduas-Dedekind (2020), a parameterisation of sea spray aerosols containing biogenic material from Wilson et al. (2015) and a parameterisation of lignin based on lignin solutions of 20 mg C L$^{-1}$ from Miller et al. (2021). The indicated solution concentrations refer to the carbon content of each solution.





We also observed that the $n_\mathrm{m}$ of the Snomax solutions did not align well as a function of temperature. Instead, we observed distinct increases in $n_\mathrm{m}$ with decreasing Snomax concentrations (Fig. 7a). This stepped profile in the Snomax freezing activity indicates that the formation of larger aggregates at higher Snomax concentrations is impacting its ice-nucleating activity. As

the Snomax concentration decreases, the larger aggregates break up, leading to a solution with a larger quantity of less active entities, which overall increases the ice-nucleating activity. Similarly to lignin, the larger aggregates also likely block sites for ice nucleation, so these sites are released into solution upon their dissolution.

We also examined the $n_\mathrm{m}$ of the soil extracts (Fig. 7b). The dilution series for the soil extracts from UBC and UoL farms align to within 2°C as a function of temperature for $n_\mathrm{m}$. However, for the Rothamsted sample, the dilution series was somewhat

aligned as a function of temperature, except for the last dilution ($0.0021\,\mathrm{mgC\,L^{-1}}$) which is up to an order of magnitude higher than the rest of the dilution series. This observation may also support the aggregation of macromolecules such as lignin or proteins of bacterial origin, which block ice-active sites at higher soil concentrations (Hartmann et al., 2022; Lukas et al., 2020, 2022).

We observed a large spread in the normalised freezing spectra of our soil extract solutions (Fig. 7b). This observation

suggests that the soil's organic carbon content cannot explain its ice-nucleating activity. Similar work has also shown that the ice-nucleating ability of agricultural soils does not correlate directly with the dissolved organic content (O'Sullivan et al., 2014). Work by Conen et al. (2011) analysed the ice-nucleating activity of agricultural soils from Mongolia, Germany, Hungary, and Yakutia. They found a variation of two orders of magnitude in the $n_\mathrm{m}$ of the different soil samples, but this variation could also not be explained by the organic carbon content of the soils (Conen et al., 2011).

The $n_\mathrm{m}$ for the lignin and Snomax solutions were compared with the $n_\mathrm{m}$ of the soil extract solutions to analyse the ice-nucleating activity and relative contributions from lignin and ice-active proteins and compared to Wilson et al. (2015) and Miller et al. (2021) (Fig. 7c). The majority of the ice-nucleating activity of the soil extracts is greater than that of lignin solutions of 20 mgC L$^{-1}$ (Miller et al., 2021) but varies due to the spread of our data. For example, at -15°C, the $n_\mathrm{m}$ of the Rothamsted sample is within about 1 order of magnitude of lignin, whereas, the $n_\mathrm{m}$ of the UoL sample is a factor of 10 to 1000

greater than lignin. This observation suggests that lignin's potential contribution to the investigated soils' overall ice-nucleating activity varies across different samples. In contrast, the ice-nucleating activity of the extracted soil solutions is lower than that of the sea surface microlayer (Fig. 7c).

The $n_\mathrm{m}$ of the soil extracts was compared with the Snomax $n_\mathrm{m}$ values. The large increase in $n_\mathrm{m}$, from $0.2\,\mathrm{mgC\,L^{-1}}$ to $100\,\mathrm{mgC\,L^{-1}}$ in the UoL Farm samples occurring at -6°C correlates with a sharp increase in ice-nucleating activity in the

Snomax data. This increase in ice-nucleating activity was also observed in the UBC soil extract solutions, as the $n_\mathrm{m}$ increased by two orders of magnitude at around -6°C. This correlating increase in $n_\mathrm{m}$ suggests that proteinaceous INMs are contributing to the ice-nucleating activity of the UoL and UBC soil extracts at these high freezing temperatures. However, smaller concentrations are present in the soil extracts compared with our pure, filtered Snomax solutions. Additionally, the high ice-nucleating activity of the soil extracts was observed even though the soil extract solutions were filtered to $0.22\,\mu\mathrm{m}$, consistent with other

studies that have also shown ice-nucleating activities remain high despite filtration (Pummer et al., 2012; O'Sullivan et al.,





2015). These two observations suggest that biogenic macromolecules contribute significantly to the ice-nucleating ability of all the soil extracts examined here.

## 4 Conclusions

This study investigated the relative contribution of surfactant macromolecules to the ice-nucleating activity of two proxies for
soil macromolecules, lignin and Snomax. We observed a correlation between the two solutions' surface tension reduction and freezing temperature. In addition, we used a high-speed cryo-microscope to reveal that ice nucleation by lignin preferentially occurs at the air-water interface of droplets in a similar manner to that previously reported for proteins from Snomax (Bieber and Borduas-Dedekind, 2024). These observations indicate that surfactants are aggregating at the air-water interface of the droplets, creating a template for ice nucleation. Similarly, we investigated surfactant macromolecules as ice-nucleating particles in three
soil samples collected in the UK and Canada. We found little reduction in surface tension in the soil extracts despite high ice-nucleating activities, indicating no clear relationship between the surface tension reduction and the ice-nucleating ability of these soils. This observation indicates that surface active macromolecules did not dominate ice-nucleating activity in these soils or that the ice-nucleating entities preferentially clustered with other components of the complex soil extract. The latter is consistent with observations by Bieber and Borduas-Dedekind (2024) who showed that in the presence of cell fragments
ice nucleation proteins had an affinity for the surfaces of those fragments as well as the air-water interface. Nevertheless, we demonstrate that surfactant properties can correlate with ice-nucleating properties, though only for soil proxies, not soil extracts. Hence, we cannot use surface tension to predict ice-nucleating activity, but it does provide a diagnostic tool that can be used to help understand the mechanisms of nucleation.

Furthermore, we found that the mass of total organic carbon in the sample alone does not allow for the prediction of its
ice-nucleating ability, which is rather a function of the complex composition of the soil samples. Previous work has also shown that some soils contain more ice-active material per mass than others (O'Sullivan et al., 2014). Heat treatments resulted in a loss of ice-nucleating activity at freezing temperatures above -10°C, indicating the importance of ice-active proteins to the ice-nucleating ability of our soil extracts. However, the overall reduction in ice-nucleating activity after heating was minor relative to reported reductions in the activity of Snomax or lichens (Daily et al., 2022), suggesting that a large proportion of the
ice-nucleating activity was attributable to heat-stable compounds such as lignin, cellulose or other polysacharides, as has been shown in previous work (Borduas-Dedekind et al., 2019; Steinke et al., 2020; Chen et al., 2021; Daily et al., 2022). However, the relative contribution of the different components remains unclear, and further study is required to better understand the mechanisms of ice nucleation in agricultural soil dust.

*Data availability.* Raw data from the FINC and tensiometer analysis are presented in a spreadsheet available alongside this manuscript. The
data are separated into different sheets within the spreadsheet. The first two sheets contain the FINC and tensiometer measurements from the



undiluted soil proxies and soil extract solutions. We then have tabs containing the measurements from the dilution series for each sample, and the heat treatment of our UBC and UoL farms.

*Author contributions.* KAT, BJM and NBD designed the research. KAT, PB, AJM, and NL made measurements and analysed the data. KAT and NBD wrote the manuscript with contributions from all authors.

*Competing interests.* The authors declare no competing interests.

*Acknowledgements.* We acknowledge financial assistance from the Engineering and Physical Sciences Research Council (EPSRC) (EP/S023593/1), the Natural Environment Research Council (NERC) (NE/V019740/1, NE/T00648X/1) and Go Global Link, Mitacs, University of British Columbia. We also thank Rachel Gasior for helping with the TOC analysis at the University of Leeds. We thank Tim Carter at UBC Farms, Jarrod Benson at UoL Farm, and Jon West at Rothamsted Research for helping with the soil sampling. The work at UBC was carried out on
the traditional, ancestral and unceded territory of the Musqueam people.



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
