# Peer review of "The role of surface-active macromolecules in the ice nucleating ability of lignin, Snomax, and agricultural soil extracts"

_EGUsphere, 2024_

## Author Response (AR1)

**Author Response to Reviews of**

**The role of surface-active macromolecules in the ice nucleating ability of lignin, Snomax, and agricultural soil extracts**

Kathleen A. Thompson, Paul Bieber, Anna J. Miller, Benjamin J. Murray and Nadine Borduas-Dedekind
*Atmospheric Chemistry and Physics,* `doi:10.5194`
* * *
RC: *Reviewer Comment*,    AR: *Author Response*,    ☐ Manuscript text

**1. Reviewer #1**

AR: *We thank the reviewer for taking the time to write constructive comments.*

RC: ***The study presented in the manuscript was motivated by the hypothesis that hydrophobic interfaces might have an influence on how organic macromolecules nucleate ice. To test this hypothesis, dilutions of lignin and snomax were tested for surface tension and ice nucleation properties. Increasing concentrations of these compounds in water resulted in decreasing surface tension and an increasing number of ice active sites. In contrast, soil extracts were reduced by dilution in the number of ice active sites with little unsystematic changes in surface tension. All laboratory work was done with great attention to detail. Overall, the study is clearly written and its results offer new insights, as far as the comparison of lignin and snomax with soil extracts is concerned. Regarding the correlation between surface tension and ice active sites observed in dilutions of lignin and snomax, I wonder whether this is not a foregone conclusion. The substances tested were already known before the experiment to have ice nucleating and surfactant properties at the same time. That both properties are less expressed at higher dilution levels is revealing the obvious. More interesting would be to have a closer look and discuss trends in T50 within dilution ranges where surface tension did not change (lignin $<10^2$ mgC/L; snomax $<10$ mgC/L), but T50 tended to increase with increasing concentration of C.***

AR: *The results for lignin and Snomax were included in part to illustrate that we can observe concurrent changes in freezing temperature and surface tension. Since the soil extracts did not follow our expected trend, it became important to show that the expected trend did indeed occur for lignin and Snomax. We may not have a clear explanation as to why the surface tension was not a good predictor of ice nucleating ability of the extracts but we wanted to showcase that it was indeed true for lignin and Snomax. In other words, we need to establish the obvious (as the reviewer calls it) to make the comparison with soil extracts. Although we appreciate the reviewer's comment that the ice nucleation and surfactant properties of lignin and Snomax were known prior to our work, this study is the first time that these two properties have been closely examined together.*

RC: ***Fig 2: Considering the counting error, is the number of freezing events at AWI significantly different from that in other compartments? If I counted correctly, 11 events occurred at the AWI, 3 in the neighbouring shell, and 6 in each of the three inner compartments. The difference between counts in AWI and in the three inner compartments does not appear to be significant because these low numbers of counts are associated with a relatively large uncertainty. Their standard deviation is roughly equivalent to the square root of counts. In identical replications of this experiment, AWI counts might well turn out to be the same***

***(11 - $\sqrt{11}$) as in inner compartments (6 + $\sqrt{6}$).***

AR: *It is correct that the fraction of freezing events in the outermost volume (11 out of 32 equals 34 %) was close to expected 20 % for volume freezing behavior when considering the uncertainties. However, the division of the volume in 5 equivalent parts was mainly motivated by the Bieber and Borduas-Dedekind (2024) paper, where the freezing was more pronounced at the air-water interface for the Snomax samples. Here, lignin showed only weak affinity to the AWI and the division into 5 parts might hide this affinity due to the low numbers of observations. To better demonstrate that lignin showed indeed some affinity for the AWI, we have changed the figure in the main text (see Fig. 1 here) and moved the target plots to the SI. Now the reader can directly compare the fraction of frozen droplets versus the location of freezing in the droplet ($r^2$). Clearly, the affinity of lignin (purple curve) to the AWI in the sector between $r^2$ = 0.8 to 1 becomes more obvious, as the curve leans towards the right-hand side (AWI affinity). We updated the methodology and modified the text in the results section accordingly:*

> We used a high-speed camera on top of a cryo-microscope to locate the onset freezing in single droplet experiments Bieber and Borduas-Dedekind (2024). This technique identified previously that the onset freezing location of Snomax samples statistically favoured the air-water interface (AWI) (Bieber and Borduas-Dedekind, 2024). In fact, Bieber and Borduas-Dedekind (2024) found the onset freezing of a filtered Snomax sample to be at the outermost fifth of the droplet in 59% of the 32 experiments (Fig. S6b). The snowmax results from Bieber and Borduas-Dedekind (2024) are shown in (Fig. 2) where we have plotted the fraction of droplets frozen against the location ($r^2$) in the cylindrical droplet. If the nucleation occurs randomly anywhere in the volume of the droplet then we expect the fraction frozen points to fall on the straight line. This is illustrated by birch pollen washing water, where the macromolecules responsible for ice nucleation are dispersed throughout the volume of the droplet. In contrast, the ice nucleating surfactant and filtered snowmax solution deviate strongly from the 1:1 line with a strong preference for freezing close to the interface ($r^2 > 0.8$). Lignin exhibits intermediate behavior, with a deviation from the 1:1 line above $r^2 = 0.6$). Nucleation at $r^2 > 0.8$ (i.e. close to the AWI) occurredin 34 % of the 32 measurements (Fig. S6a). This result indicates that there is enhanced ice-nucleating activity of lignin at the AWI (Fig. 2a), however, it is less pronounced compared to the Snomax samples (Fig. 2b).

RC: ***Line 270: replace 'step-wide' with 'step-wise'***

AR: *Thank you for pointing this out. We have corrected this mistake in the revised manuscript.*

RC: ***Line 347 replace 'heat-label' with 'heat-labile'***

AR: *Thank you for pointing this out. We have corrected this mistake in the revised manuscript.*

RC: ***Figure 6: Consider using the same unit in the y-axis label as in Figure 7 ([(mg C)$^{-1}$] instead of [mg C$^{-1}$]).***

AR: *We agree and have made the change.*

[Figure]

Figure 1: **Fraction of frozen droplets ($f_f$) plotted against the onset freezing locations ($r^2$) for cylindrical droplets squeezed between two protein-repellent glass slides**. Lignin ($5000\,\mathrm{mgC\,L^{-1}}$) is plotted and compared with filtered Snomax, Docosanol and *P. syringae* cells adapted from Bieber and Borduas-Dedekind (2024). The mean freezing temperatures were -20.6 °C for lignin and -6.3 °C for the filtered Snomax sample, which is in the range of the investigated concentrations. Note that the droplet volume was 0.5 $\mu$L for these measurements.

**2. Reviewer #2**

**RC:** *This manuscript is well-written in general. While this reviewer is not 100% sure about the atmospheric implication of the study outcomes, this paper discusses an important negative result regarding potential ambient ice nucleation – surface active macromolecules do not dominate INPs in bulk soil samples. The study topic is relevant to the journal scope of ACP. This reviewer supports the publication of this paper in AMT after addressing several questions below.*

AR: *Thank you for supporting the publication of our paper, we will address your questions below.*

**2.1. Questions**

**RC:** *(1) Can surface tension and micelle formation of heated soil extracts be measured? If yes, can they be reported in this manuscript? Doing so might clarify the next question.*

AR: *We did measure the surface tension of the heated soil extracts. We report the results of this in Section 3.2.3.*

**RC:** *(2) Can heat-treated soil extracts show what the authors find in lignin (an increase in conc is proportional to an increase in IN activity and a reciprocal decrease in surface tension)? Adding discussions regarding surface properties of heat-treated soil suspensions at this stage might be worth to shed light on the relative importance of surface active MMs to IN active protein in soil samples. Recently, as the authors discuss in L35-37, some heat-sensitive proteins/enzymes commonly found in soil, such as Rubisco, have been found*

*to be IN active. Heat treatment presumably removes these proteins, and the remaining soil organics can show lignin-like surface properties?*

AR:   *Our analysis revealed that heating the soil extracts had no observable effect on the surface tension, despite some clear reduction in ice-nucleating activity. We will add to our discussion of this to comment on why we think we observe this instead of seeing that the remaining soil organics after heating show lignin-like surface properties:*

> Surface tension measurements were also taken before and after heating the soil extract solutions (see Table S2). We anticipated that removing heat-sensitive INMs, such as proteins, may have allowed us to see the relationship between surface tension and ice-nucleating activity, as observed in our lignin solutions (Fig. 4a). However, for both the UoL Farm and UBC Farm samples, we observed little change in the surface tension reduction after heating (Table S2). This observation indicates that any surfactant molecules within the extract solutions were not degraded when the solutions were heated, consistent with our conclusion that the molecules responsible for ice nucleation within our soil extracts were not surface active in nature.

**2.2. Comments**

RC:   *P1L4: Why the impact needs to be on only 'local' cloud formation?*

AR:   *They don't, we have removed the word 'local'.*

RC:   *P4L119-121: How long the tube was opened to sample ambient air? Quantitatively clarifying the blank handling protocol in the manuscript will be helpful for the reader and whoever may follow up on your study.*

AR:   *We have added the following to clarify this:*

> Additionally, at each location, a clean 50 mL centrifuge tube was opened and exposed to ambient air for 20 to 30 seconds, to examine the contamination of ice-nucleating particles from handling the samples (i.e. a handling blank).

RC:   *P5L132: Not SA water?*

AR:   *Correct. MilliQ water was used for the extracted soil solutions since the extraction process was partially developed in the Leeds lab, where MilliQ was used as standard. So, we used MilliQ to keep the process consistent across the two labs. A clarifying sentence has been added to the following:*

> This extraction technique was developed across the labs in both Leeds and UBC, so MilliQ water was used instead of SA water to maintain consistency across the two labs. First, 40 mL of MilliQ water  was added to 40 g of the soil samples to create a $10^6 \, \mathrm{mg \, L^{-1}}$ concentration suspension.

**References**

Bieber, P. and Borduas-Dedekind, N.: High-speed cryo-microscopy reveals that ice-nucleating proteins of *Pseudomonas syringae* trigger freezing at hydrophobic interfaces, Science Advances, 10, eadn6606, 10.519410.1126/sciadv.adn6606, 2024.

This document was generated with a layout template provided by Martin Schrön (`github.com/mschroen/review_response_letter`).